# Metformin Improves Burn Wound Healing by Modulating Microenvironmental Fibroblasts and Macrophages

**DOI:** 10.3390/cells11244094

**Published:** 2022-12-16

**Authors:** Liangliang Shi, Zhengying Jiang, Jiaqi Li, Huan Lin, Bin Xu, Xincheng Liao, Zhonghua Fu, Haiyong Ao, Guanghua Guo, Mingzhuo Liu

**Affiliations:** 1Medical Center of Burn Plastic and Wound Repair, The First Affiliated Hospital of Nanchang University, Nanchang 330006, China; 2Jiangxi Key Laboratory of Nanobiomaterials, Institute of Advanced Materials, East China Jiaotong University, Nanchang 330013, China

**Keywords:** burn, metformin, AMPK, mTOR, collagen I, NF-κB, co-culture

## Abstract

Metformin, a biguanide, exerts different functions through various signaling pathways. In order to investigate the function and mechanism of metformin in burn wounds, we established burn rat models, subcutaneously injected metformin to treat the wounds, and observed the morphologies and the expression of collagen I, collagen III, fibronectin, and pro-inflammatory markers. In vitro experiments were performed to investigate the effects of metformin on the proliferation, migration, and collagen I synthesis of the mouse embryonic fibroblast (NIH 3T3) cell line and on the proliferation, apoptosis, and immune response of the mouse mononuclear macrophage (RAW 264.7) cell line. Finally, we studied the regulatory effects of metformin on a co-culture of RAW 264.7/NIH 3T3 cells. We found that 100 mM of metformin reduced dermal thickness, collagen I deposition, and mRNA expression of IL1β and CCL2 in rat burn wounds. In vitro experiments revealed that metformin inhibited the proliferation of NIH 3T3 and RAW 264.7 cells. Metformin attenuated NIH 3T3 cell migration via the AMPK/mTOR pathway and attenuated collagen I synthesis through the TGFβ1/Smad3 pathway. Metformin inhibited the apoptosis of RAW 264.7 cells induced by 10 μg/mL LPS. Metformin downregulated the mRNA expression of IL1β and CCL2 in RAW 264.7 cells under 1 μg/mL LPS induction by inhibiting NF-κB p65 phosphorylation. In a RAW 264.7/NIH 3T3 co-culture, metformin attenuated collagen I synthesis in NIH 3T3 cells by inhibiting RAW 264.7 paracrine secretion of TGF-β1. This provides new evidence related to the development of metformin for potentially improving burn wound healing.

## 1. Introduction

Burns are damage to the skin or other tissues caused by heat, electricity, chemicals, etc. Although the incidence of burns is decreasing year by year in developed countries, there are still a large number of patients in developing countries who suffer from burns for various reasons every year, with minor disfigurement, limb deformity, or death, indicating that burns seriously endanger public health [1]. Deep second-degree burns are classified as deep dermal burns with residual skin attachments and dull sensations, requiring 3 to 4 weeks of healing, and result in the formation of scars. Deep second-degree burns tend to deepen scarring, because burn wounds are often associated with infection, wound compression, residual necrotic tissue, and local inflammatory imbalances that deepen the wound, leading to slow healing and the formation of hypertrophic scars [2]. Post-burn hyperplastic scarring and subsequent possible scar contracture are known to be two important problems affecting patients, and their management directly affects the level of post-burn functional recovery [3]. Conventional treatment modalities for hypertrophic scarring following deep second-degree burns include medication, compression therapy [4], radiotherapy, laser therapy, gene therapy, immunotherapy, and surgery [5]. However, the mechanism of occurrence of hypertrophic scarring has not been fully elucidated. At present, clinical treatment is complex, there is no specific drug, and the recurrence rate is high. Therefore, it is necessary to find a drug with wide applications and which yields good results. 

However, because wound repair is a result of the combined action of multiple mechanisms in multiple cells, it has been proposed to use the wound microenvironment to simulate and study the real wound healing process. Multiple studies have demonstrated the critical roles of fibrosis and inflammation in scar formation. Additionally, fibroblasts and macrophages play an important role in fibrosis and inflammation. Hence, strategies that target fibroblast and macrophage signaling pathways may provide promising opportunities to protect against hypertrophic scarring.

Metformin has been used as a first-line agent in treatments for type 2 diabetes (T2DM) for over 60 years [6]. In 2020, the American Diabetes Association Diabetes Guidelines recommended the drug as a first-line treatment for T2DM [7]. The mechanism of glycemic control by metformin remains an area of active research. There are currently three proposed mechanisms: (1) complex I inhibition, where complex I acquires two electrons from the reducing force NADH obtained from the TCA cycle, which are passed to coenzyme Q via iron–sulfur proteins. In this process, complex I provides energy for gluconeogenesis. Metformin alters the energy charge of hepatic adenine nucleotides, leading to complex I inhibition, which reduces ATP/ADP and ATP/AMP ratios [8]; (2) AMPK activation, because metformin is proposed to activate AMPK by inhibiting complex I and reducing ATP/ADP and ATP/AMP ratios. The activation of AMPK leads to the downregulation of gluconeogenic gene transcription, which reduces hepatic glucose production (HGP) and the phosphorylation of ACC1 and ACC2, thereby reducing adipogenesis and promoting hepatic mitochondrial oxidation [9], resulting in reduced hepatic diacylglycerol content and increased hepatic insulin sensitivity [10]; (3) an increased cytosolic redox state—a novel mechanism that has been suggested in which the glycemic control effect of metformin is due to the inhibition of hepatic GDP2 activity and thus the enhancement of cell membrane redox [11].

In recent years, a number of studies have confirmed that metformin has therapeutic effects, such as anti-aging effects, cardiovascular protection, PCOS improvement, anti-tumoral properties, autism treatment, attenuation of silica-induced pulmonary fibrosis, smoking cessation, anti-inflammatory effects, and reversal of cognitive impairment [12,13,14,15,16,17,18,19,20]. Regarding these effects, although the anti-aging properties, reversal of pulmonary fibrosis, and anti-inflammatory efficacy of metformin have been studied and proven to act through the mitochondria, the mechanisms for other disease-specific details remain largely unclear. In vitro and in vivo assays have shown that metformin promotes the formation of the AMPKαβγ heterotrimeric complex, which thus activates AMPK, possibly explaining the pleiotropic effect of metformin [21]. Nevertheless, this has yet to be detailed and dissected.

Metformin has been shown to attenuate fibrosis in multiple organs, such as the liver, lungs, kidneys, and myocardium [22,23,24,25]; however, whether and how it affects the proliferation of post-burn wounds and its mechanisms have not been studied.

We aimed to observe the effects of different drug concentrations on burn wounds and select optimal drug concentrations in our experimental studies; observe the effects of metformin modulating AMPK signaling on fibroblasts and macrophages in vitro; and explore the possible regulatory mechanisms and how cellular interactions in local microenvironments affect cytokine secretion and expression. Through these experiments, we sought to reveal the mechanism of AMPK in improving the healing of burn wounds and provide a new approach for improving burn wound healing.

## 2. Materials and Methods

### 2.1. Animal Experiments

In total, 12 healthy SD rats, 6–8 weeks old and with body weights in the range of 160–180 g, were purchased from the Animal Center, Jiangxi University of Chinese Medicine, Nanchang, China, and fed in a clean environment. The rats were anesthetized with intraperitoneal chloral hydrate injection. The rats’ backs were shaved to expose the skin. The back skin was burned with boiling water at 95 ℃ for 8 s in a relatively symmetrical manner, with a diameter of 2 cm [26]. The margins of each burn wound in our in vivo experiment were spaced at least 3 cm apart. Each burn wound on the dorsal surface of all rats was subcutaneously injected with 1 mL of 10 mM PBS solution or different concentrations of metformin (1 mM, 10 mM, 100 mM) solution. Five groups of tissues were investigated: wounds treated with 1 mM metformin (Met 1 mM), wounds treated with 10 mM metformin (Met 10 mM), wounds treated with 100 mM metformin (Met 100 mM), PBS-treated wounds (Met 0 mM) as a positive control, and normal skin tissue as a negative control (Figure 2A,B). We changed the dressings every two days with dry sterile gauze and disinfected the wounds with iodophor. To try and avoid excessive irritation of the wounds, we did not remove scabs, even if they were loose. For intermittent applications, agents were administered every other day per week, followed by one week without treatment. Then, the rats were sacrificed and the skin tissues were isolated. The wound bed biopsy was divided equally into parts for paraffin-embedded samples, frozen sections, and protein extractions. 

### 2.2. Cell Culture

The NIH 3T3 and RAW 264.7 cells were maintained in DMEM (Gibco, Waltham, MA, USA) supplemented with 10% fetal bovine serum (Gibco, Waltham, MA, USA) and 1% penicillin/streptomycin (100 U/mL penicillin and 100 µg/mL streptomycin; Solarbio, Beijing, China). The cell lines were incubated in a humidified atmosphere with 5% CO_2_ at 37 °C. Logarithmic growth cells were used for the experiment. Cells were sub-cultured at a ratio of 4 × 10^6^ cells/flask and in 6-well cell culture plates at 5 × 10^5^ cells/well. Culture media were freshly changed every day.

### 2.3. Hematoxylin and Eosin Staining

Rat skin tissue was routinely fixed, sliced, and stained with hematoxylin and eosin (HE) for histological investigations. 

### 2.4. Masson Staining

Masson staining, which distinguishes between collagen fibers and muscle fibers, rendering collagen fibers blue (aniline blue) and muscle fibers red (lixin red and acidic magenta), is the conventional method of connective tissue dyeing. The rat skins were quickly removed under anesthesia and blood was cleaned with PBS. The skins were fixed with 4% paraformaldehyde. Samples were embedded in paraffin and cut into 3 mm-thick slices. After staining with Masson’s trichrome stain, the slices were observed under a light microscope (CKX41, Olympus, Tokyo, Japan), and the sections were imaged using a digital microscope. Samples were observed at 400× magnification and three horizons were tested in each sample. The volume fraction of collagen (CVF) in the skin tissues was analyzed with ImageJ/Fiji software 1.53v (NIH, Bethesda, MD, USA).

### 2.5. Picrosirius Red Staining

Picrosirius red staining is a straightforward and sensitive method for identifying collagen fibril networks in tissue sections. Rat skin tissue was routinely fixed, sectioned, and stained with Picrosirius red (Serviecbio, Wuhan, China). Four collagen fiber types (I, II, III, IV) were shown under polarization microscopy (Nikon, Tokyo, Japan) based on collagen birefringence and staining. Polarization microscopy revealed type III collagen fibers with low birefringence and appeared as fine green fibers. The areal percent of collagen III was measured with ImageJ/Fiji software 1.53v (NIH, Bethesda, MD, USA).

### 2.6. Immunohistochemical Staining

Briefly, paraffin-embedded rat skin was routinely de-paraffinized and re-hydrated. The samples were incubated with diluted primary antibodies at 4 °C overnight and then processed with the iVision™ Poly-HRP goat anti-mouse/rabbit secondary antibody reagent (Talentbiomedical, Xiamen, China, DD13) and ABC complexes. Finally, samples were reacted with a DAB kit (Servicebio, Wuhan, China, G1212-200T). Images were acquired with an optical microscope (CKX41, Olympus, Tokyo, Japan) and analyzed for IOD in each view.

The primary antibodies used were as follows:

Anti-Collagen III (Proteintech, Rosemont, IL, USA, 22734-1-AP, 1:500);

Anti-Fibronectin (Proteintech, Rosemont, IL, USA, 15613-1-AP, 1:500).

### 2.7. Cell Viability Assay 

Cell proliferation was detected using the cell counting kit-8 (CCK8), following the manufacturer’s instructions (GLPBIO, Montclair, CA, USA). Cells were incubated for 24 h in media containing different concentrations of metformin (0.1 mM, 0.5 mM, 1 mM, 5 mM, 10 mM, 20 mM); 100 μL of CCK8 reagent was added to each well and left for 1 h. The OD of each well was measured at 450 nm using a fully automated microplate reader, and the values were recorded for subsequent data analysis. IC50 values and 95% confidence intervals were calculated (Table 1). 

### 2.8. EdU Staining 

As a thymidine nucleoside analog, EdU (5-Ethynyl-2′-deoxyuridine) can be used to replicate DNA rather than the thymidine cell proliferation phase. The proliferation ability of cells can be detected quickly and accurately through the reaction of EdU with fluorescent dyes. Transfected NIH 3T3 cells were seeded onto 12-well plates at a density of 5 × 10^3^ cells/well and cultured to normal adherent growth. The Cell-Light EdU Apollo567 in Vitro Kit (RiboBio, Guangzhou, China) was used to incubate cells with EdU (50 μM) for 2 h, in accordance with the manufacturer’s instructions. Cells were fixed at room temperature with 4% paraformaldehyde formaldehyde for 30 min, then dyed with Apollo for 30 min. After 30 min of nuclear staining, cells were observed under an inverted fluorescence microscope (Carl Zeiss AG, Jena, Germany). EdU-positive cells were counted as percentages of all EdU-positive cells randomly captured in each well.

### 2.9. Transwell Migration Assay

The cell density was adjusted to 6 × 10^5^ cells/mL by microcount using serum-free DMEM resuscitation. Approximately 200 μL of cell suspension was added to each well in the 24-well plates with a cell count of approximately 1.2 × 10^5^ cells/mL. The lower chamber of each well was infused with 600 μL of complete media, and the upper chamber was carefully impregnated with sterile forceps to avoid the formation of air bubbles between the upper chamber and lower chamber, affecting the observation results. Finally, the 24-well plates were returned to the CO_2_ cell incubator and incubated for 24 h. The media from each of the upper and lower chambers in the 24-well plates were then flushed twice using 500 μL of PBS. Each lower chamber was fixed at 400 μL, 4% paraformaldehyde, for 15–20 min. Subsequently, 500 μL of PBS was added to each upper and lower compartment and these were washed twice to remove the formaldehyde. Then, 400 mL of 1% crystalline purple was added to each chamber and stained for 30 min. Cells stained with 1% crystal violet were washed 3 times with natural water and air-dried for 12–24 h in a ventilated place. Cell migration was observed under 5 random fields of vision under a microscope.

### 2.10. Cell Apoptosis Assay

RAW 264.7 cells were seeded in 6-well plates at a density of 1 × 10^6^/well. Pretreatment with metformin at different concentrations for 18 h was followed by further intervention with or without the addition of 10 μg/mL LPS for 6 h. Following EDTA-free trypsin digestion, cells were harvested and suspended in binding buffer, followed by mixing with Annexin V-FITC and PI. This was performed at room temperature, with protection from light, for 10 min, followed by observation and detection by flow cytometry (BD, Franklin Lakes, NJ, USA). The results were analyzed using FlowJo 7.0 (BD, Franklin Lakes, NJ, USA).

### 2.11. Cell Co-Culture 

Cell co-culture analysis was performed in Transwell chambers (Corning, NY, USA) containing 6.5 mm diameter polycarbonate filters (0.4 μm pore). NIH 3T3 cells at a seeding density of 1 × 10^6^/well in the lower chambers and RAW 264.7 cells with a seeding density of 0.3 × 10^6^/well in the upper chambers were defined as the RAW 264.7/NIH 3T3 group. Both the upper and lower chambers were seeded with NIH 3T3 cells, defined as the NIH 3T3/NIH 3T3 group, as controls. Each group was divided into 3 sub-groups by interventions, such as LPS (1 μg/mL) and Met (10 mM). After a total of 48 h of co-culture, the collagen I expression was evaluated by Western blotting of NIH 3T3 cells in the lower chambers. The expression of TGF-β1 in the supernatant of each group was determined by ELISA.

### 2.12. siRNA Assay

AMPK siRNA was synthesized by Gene Pharma (Shanghai, China). Following the manufacturer’s instructions, cells were instantaneously transfected with Lipofectamine 2000 reagent (Invitrogen, ThermoFisher, Waltham, MA, USA). NIH 3T3 cells were cultured in six-well plates in DMEM (2 mL per well) without antibiotics or FBS, then transfected with 5 μL of 20 μM siRNA per well using 5 μL Lipofectamine 2000 reagent. The steps were as follows: each siRNA (5 μL) was diluted in 100 μL of OPTIMEM I reduced serum medium; 5 μL Lipofectamine was diluted in 100 μL of OPTIMEM I reduced serum medium; the transfection complexes (200 μL) were placed in 6-well culture plates and incubated at 37 °C in 5% CO_2_ for 24 h. AMPK activity, assessed by monitoring the phosphorylation of AMPK at Thr172, was significantly downregulated when NIH 3T3 cells were transfected with AMPK siRNA for 24 h (Figure 1C,D). All siRNA sequences used in this study are shown in Table 2. We chose Prakaa1-mus-1337 for the next stage of the experiment.

### 2.13. Lentiviral shRNA Assay

shRNA specific to mouse AMPK plasmids was synthesized by HanBio (Shanghai, China) (Table 3). The plasmids were transfected into NIH 3T3 cells using polybrene (Shanghai, China) to enhance virus transduction. Cell lines exhibiting the stable lentiviral knockdown of AMPK were screened stepwise with puromycin-containing medium, and the knockdown of AMPK and pAMPK was confirmed by Western blotting (Figure 1E–G). We selected shAMPK3 for further study.

### 2.14. Quantitative RT-PCR (qRT-PCR)

RNA from rat skin tissue and RAW 264.7 cells was extracted using TRIZOL and reverse-transcribed into cDNA using PrimeScript RT Master Mix (ABI, Waltham, MA, USA), in accordance with the manufacturer’s protocol (Tiangen Biotech, Beijing, China). Primer sequences are shown in Table 4 and Table 5. Gene expression was normalized for GAPDH using the ΔΔCT method.

### 2.15. Western Blotting

Total proteins isolated from treated NIH 3T3 cells, RAW 264.7 cells, and rat skin tissues were separated by gels using SDS-PAGE. The proteins were transferred to PVDF membranes and sealed at room temperature with 5% skim milk for 1 h. The PVDF membranes were incubated overnight at 4 °C with appropriate primary antibodies. The membranes were washed 3 times, 10 min each time, with 1 × TBST buffer, then incubated at room temperature with corresponding HRP conjugated secondary antibodies for 1 h. Each time, PVDF membranes were washed 3 times with 1 × TBST for 10 min followed by treatment with enhanced chemiluminescence reagent (ECL) (BioSience, Shanghai, China). Images were obtained using the Molecular Manager® ChemiDoc™ XRS+ Imaging System (Bio-Rad, Hercules, CA, USA) and quantified using Image Lab analysis software version 4.1 (Bio-Rad, Hercules, CA, USA).

The primary antibodies used were as follows:Anti-AMPKα1 + α2 antibody (Abcam, ab207442, 1:1000);Anti-AMPKα1 (phospho T183) + α2 (phospho T172) antibody (Abcam, ab133448, 1:5000);Anti-TGF-β1 antibody (Abcam, ab215715, 1:1000);Anti-Smad3 antibody (Abcam, ab208182, 1:1000);Anti-Smad3 (phospho S423 + S425) antibody (Abcam, ab52903, 1:2000);Anti-Collagen I antibody (Abcam, ab260043, 1:1000);Anti-mTOR antibody (Abcam, ab134903, 1:10000);Anti-mTOR (phospho S2448) antibody (Abcam, ab109268, 1:5000);Anti-AKT1 + AKT2 + AKT3 antibody (Abcam, ab179463, 1:10000);Anti-AKT1 + AKT2 + AKT3 (phospho S472 + S473 + S474) antibody (Abcam, ab192623, 1:1000);Anti-MMP2 antibody (Abcam, ab181286, 1:1000);Anti-MMP9 antibody (Abcam, ab228402, 1:1000);Anti-NF-κB p65 antibody (Abcam, ab32536, 1:5000);Anti-NF-κB p65 (phospho S536) antibody (Abcam, ab76302, 1:000);Anti-GAPDH antibody (ZSGB-Bio, 1:1000).

### 2.16. ELISA 

The culture supernatant collected was collected. TGF-β1, IL-1β, and CCL2 were stored at −80 °C until required for each assay. Levels of TGF-β1 (detection range: 15.6–1000 pg/mL; sensitivity < 1 pg/mL of recombinant mouse TGF-β1), IL-1β (detection range: 15.6–1000 pg/mL; sensitivity < 1 ng/mL of recombinant mouse IL-1β), and CCL2 (detection range: 15.6–1000 pg/mL; sensitivity < 0.5 ng/mL of recombinant mouse CCL2) in the culture supernatant (samples per group) were determined using a commercial ELISA kit (BOSTER, Wuhan, China). Concentrations of cytokines, including TGF-β1, IL-1β, and CCL2, were determined by ELISAs, according to the manufacturer’s protocol. All samples were assayed for OD values at 450 nM using a microplate reader.

### 2.17. Data and Statistical Analysis

GraphPad 8.0 statistical software (GraphPad, San Diego, CA, USA) was used for all data and statistical analysis. Intergroup differences were analyzed by two-sided unpaired Student’s *t*-test or one-way analysis of variance (ANOVA) and Tukey’s post hoc test following a normal distribution test. The Student’s *t*-test and ANOVA data were used for statistical significance analyses; *p* values < 0.05 indicated significant differences.

## 3. Results

### 3.1. Metformin Inhibits Collagen I Deposition In Vivo

After the burn model was established, subcutaneous drug injection was performed every other day for seven days (Figure 2A). Two weeks after the model was established, the wounds were all closed with no occurrences of wound infection or delayed closure inter/intra-individually. The skin tissue was extracted: part of it was used for HE staining and Masson staining; part of it was used for protein extraction. In this study, the thickness of the dermis and the collagen volume fraction (CVF) were significantly reduced in the 100 mM MET group (Figure 2B–E), while no significant differences were found in the expression of collagen III or fibronectin (*p* > 0.05) (Figure 2H–M). Western blot analyses showed that collagen I expression in the 100 mM metformin group was lower than in the burn group (*p* < 0.001) (Figure 2F,G). This suggests that metformin attenuates the deposition of collagen I in burn wounds in rats.

### 3.2. Metformin Reduces IL1β and CCL2 mRNA Expression in Burn Wounds in Rats

On the fourth day after the model had successfully been established, 0.5 cm margins of the burn wounds were harvested. RNA was extracted for RT-PCR assays. The observed mRNA expression of IL1β and CCL2 in the skin tissues showed that levels in the metformin group were significantly lower than in the burn group (Figure 2N,O).

### 3.3. Metformin Inhibits NIH 3T3 Proliferation via the Activation of AMPK In Vitro

To investigate whether metformin inhibits cell proliferation, the CCK-8 assay was employed to evaluate its proliferation effect in NIH 3T3 cells. NIH 3T3 cell proliferation was inhibited following treatment with different concentrations of metformin in a dose-dependent manner (Figure 3A).

We observed a significant attenuation of cell proliferation 24 h after treatment with either 5 mM or 10 mM of metformin compared with the control. 

Then, we used AMPK siRNA to downregulate AMPK. The percentage of EdU-positive cells increased compared with the negative control group. We observed that the cell proliferation ability was significantly enhanced after transfection (Figure 3B,C).

### 3.4. Metformin Attenuates NIH 3T3 Migration through the AMPK/mTOR Pathway In Vitro 

We stimulated NIH 3T3 cells with metformin at different concentrations and found that the activation of AMPK in NIH 3T3 cells in the 0–10 mM range for metformin showed a dose-dependent effect (Figure 1A,B). To investigate the effects of metformin on NIH 3T3 cell migration, we investigated the effects of different concentrations of metformin on cell migration using the Transwell migration assay. After 24 h of intervention with 2.5 mM, 5 mM, and 10 mM metformin, migration cells in each group were observed and photographed (Figure 4A).

Dose-dependent differences were statistically significant (*p* < 0.001) (Figure 4B), suggesting that metformin inhibits NIH 3T3 cell migration (Figure 4B). Correspondingly, the levels of mTOR and AKT phosphorylation exhibited reverse changes compared with the levels of metformin. Moreover, we also found that levels of MMP2 and MMP9 decreased in a dose-dependent manner (Figure 4C–G).

Specific AMPK-siRNA was used to study the effect of AMPK signaling on the inhibition of cell migration by metformin in vitro. Interestingly, the knockdown of AMPK promoted the migration of NIH 3T3 cell migration compared with the NC group (Figure 5A,B); the knockdown of AMPK, the levels of mTOR and AKT phosphorylation, and MMP2 and MMP9 expression increased correspondingly (Figure 5C–G).

We further used a commercial plasmid to downregulate the expression of AMPK; the phosphorylation of mTOR and AKT, MMP2, and MMP9 was significantly increased in AMPK compared with corresponding NC groups (Figure 5H–L). Taken together, our results suggest that AMPK regulates the metformin-induced inhibitory effect on the cellular migration of NIH 3T3.

### 3.5. Metformin Downregulates the Expression of Collagen I in NIH 3T3 Cells via the Activation of AMPK

To investigate the anti-fibrotic effect of metformin in vitro, we treated NIH 3T3 cells with different concentrations of metformin for 24 h. We then measured the protein expression of TGF-β1, Smad3, pSmad3, and collagen I, and found that the levels of all three proteins (TGF-β1, pSmad3, and collagen I) in the Met group were significantly lower than in the control group (Figure 6A–D).

To further substantiate the role of AMPK in the antifibrotic activity of metformin, we used AMPK siRNA to downregulate AMPK expression. Key proteins in TGF-β/Smad3 were determined by Western blotting. The protein expressions of TGF-β1, pSmad3, and collagen I were significantly increased in AMPK siRNA treatment conditions (Figure 6E–H).

In addition to AMPK siRNA, we used a commercial plasmid to downregulate AMPK expression to explore its role in relation to the anti-fibrotic effect of metformin. However, in Figure 5I, Smad3 protein expression levels remained unchanged in all treatment conditions; in contrast, TGF-β1, p-Smad3, and collagen I were all significantly upregulated in plasmid treatment groups with AMPK knockdown compared with the corresponding negative control groups (Figure 6I–L). 

The results showed that the knockdown of AMPK significantly upregulated the TGF-β1-induced upregulation of collagen I at the protein level. AMPK cross-talk occurred via the TGF-β/Smad3 pathway in NIH 3T3 cells. 

### 3.6. Metformin Inhibits RAW 264.7 proliferation and Attenuates Apoptosis after High Concentrations of LPS Stimulation In Vitro

The proliferation of RAW 264.7 cells was inhibited following treatment with metformin in a dose-dependent manner (Figure 7A). RAW 264.7 cells pretreated with metformin at different concentrations for 24 h did not exhibit significant apoptosis (Figure 7B,C). Nevertheless, RAW 264.7 cells pretreated with different concentrations of metformin for 16 h, followed by the addition of 10 μg/mL LPS for 8 h of continuous stimulation, showed varying degrees of apoptosis (Figure 7D,E). The early apoptosis of RAW 264.7 cells after metformin pretreatment was significantly reduced.

### 3.7. Metformin Suppresses LPS-Induced Inflammation through Inhibiting the NF-κB Pathway in RAW 264.7 Cells

To study the anti-inflammatory effects of metformin on RAW 264.7 cells treated with 1 μg/mL LPS, the cells were pretreated with metformin for 18 h prior to 6 h of combined LPS treatment. The proteins were extracted and the phosphorylation levels of NF-κB in the RAW 264.7 cells were detected by Western blot assays. The IL1β and CCL2 mRNA expression levels were measured by RT-PCR after the same treatment. The results showed that metformin suppresses the NF-κB signaling pathway to regulate the immune response of RAW 264.7 cells. The protein expression of pNF-κB and mRNA expression of IL1β and CCL2 were significantly reduced in each intervention group in a dose-dependent manner compared with the LPS group (Figure 8A,B,E,F). The ELISA results showed that: amounts of secreted CCL2 increased in RAW 264.7 cells after LPS stimulation; metformin reduced CCL2 secretion in RAW 264.7 cells in a dose-dependent manner (Figure 8G); and IL1β was not detected in this experiment due to its low levels in the cell supernatant.

QNZ (EVP4593) was administered to inhibit the activation of the NF-κB signaling pathway, and metformin, an AMPK activator, was also administered to the RAW 264.7 cells. The cells were pretreated with metformin (0–10 mM) and QNZ (10 ng/mL) for 16 h and then stimulated with LPS (1 μg/mL) for 8 h. Levels of IL-1β and CCL2 expression were measured by RT-PCR, and NF-κB phosphorylation was detected by Western blot assay. Under conditions in which NF-κB was not activated, metformin could no longer inhibit the expression of the above cytokines (Figure 8C,D,H,I).

### 3.8. Metformin Leads to a Reduction in Collagen I in NIH 3T3 Cells by Inhibiting TGF-β1 Paracrine Secretion from RAW 264.7 Cells in a Co-Culture Model

We used in vitro co-culture to simulate the in vivo burn microenvironment, co-cultured NIH 3T3 and RAW 264.7 cells for 48 h (NIH 3T3 and NIH 3T3 co-culture was used as the control group), and observed the collagen I synthesis of NIH 3T3 cells after adding metformin (10 mM) with or without LPS (1 μg/mL). In the NIH 3T3/NIH 3T3 group, after the system was co-cultured with 1 μg/mL LPS for 48 h, the collagen I expression in NIH 3T3 cells in the lower chamber was significantly reduced compared with the control group, and collagen I expression in NIH 3T3 cells with the concurrent addition of metformin was significantly reduced compared with both the control and LPS groups; in the RAW 264.7/NIH 3T3 group, after the system was co-cultured with 1 μg/mL LPS for 48 h, the expression of collagen I in NIH 3T3 cells in the lower chamber was significantly increased compared with the control group, and collagen I expression in NIH 3T3 cells with the simultaneous addition of metformin was significantly decreased compared with both the control and LPS groups (Figure 9A,B).

The ELISA results showed that RAW 264.7 cells secreted high levels of TGF-β1 in response to 6 h of LPS stimulation and that this effect could be attenuated by metformin pretreatment for 18 h (Figure 9C). We then performed ELISAs using supernatants from the co-cultures, and the results showed that TGF-β1 expression was relatively low in the NIH3T3/NIH 3T3 groups, with lower levels of expression in the LPS+ Met− and LPS+ Met+ groups compared with the LPS− Met− group; the levels of TGF-β1 expression in the LPS+ Met− group and in the RAW 264.7/NIH 3T3 group were significantly increased compared with the LPS− Met− group, whereas the LPS+ Met+ group showed significantly lower TGF-β1 expression levels compared with the LPS+ Met− group (Figure 9D). These results suggest that RAW 264.7 cells induced an increase in NIH 3T3 collagen I expression through the massive paracrine secretion of TGF-β1 in response to LPS stimulation, whereas metformin was able to downregulate collagen I expression in NIH 3T3 cells by inhibiting the paracrine secretion of TGF-β1 from RAW 264.7 cells.

## 4. Discussion

The treatment of deep burn wounds remains a major clinical challenge due to the deterioration of the tissue microenvironment, including extracellular matrix loss, excessive inflammation, impaired angiogenesis, and bacterial infection. Various methods have been developed to reduce the deepening of second-degree burn wounds and to improve the quality of wound healing, such as nicorandil, hydrogen-rich saline, and astaxanthin treatment [27,28,29], which have different mechanisms of action, poor clinical results, and many limitations in their application. 

AMPK, or AMP-dependent protein kinase, is a key molecule regulating energy metabolism and is central to research in diabetes and the treatment of other diseases linked to metabolism. Different types of AMPK proteins are heterozygotes consisting of an α-catalytic subunit, a β-regulatory subunit, and γ-regulatory subgroups. Two isoforms of α-subunits (α1 and α2), two isoforms of β-subunits (β1 and β1), and three isoforms of the γ-subunit (γ1, γ2, and γ3) were expressed by different genes in humans and rodents. The binding site of threonine 172 (Thrl72) in its kinase region and its phosphorylation play an important role in regulating AMPK activity [30]. Many kinds of AMPK activators have been discovered, with different activation mechanisms, such as metformin, resveratrol, AICAR, melatonin, etc. Additionally, there have been a number of recent studies on the role of AMPK agonists in wound healing.

Metformin is a typical AMPK activator, which activates AMPK by changing the ATP/ADP and ATP/AMP ratios in both in vivo and in vitro assays, which may lead to diverse responses in different tissues [31]. It has been reported that the long-term external use of metformin or resveratrol promotes wound healing in rats and improves the epidermis, follicles, and collagen deposits, which are associated with improved vascular formation in wounds. Notably, this study further confirmed that metformin promotes wound healing by activating AMPK in aging rats and promotes wound vascularization, reverses aging, and reduces the scarring of aging skin, providing a theoretical basis for metformin as a treatment for wounds [32]. Faraz Chogan et al. designed, optimized, and prepared a three-layer nanofiber scaffold [33]. The scaffold consisted of two supportive polycaprolactone (PCL)–chitosan layers and polyvinyl alcohol (PVA)–metformin hydrochloride (metformin–HCl). Exposure to metformin significantly reduced the expression of fibrosis-related genes: TGF-β1, collagen I (Col-I), fibronectin (Fn), collagen III (Col-III), and α-SMA. The inhibition of these genes reduces scar formation but delays wound healing. However, the introduction of this slow-release engineered scaffold has the dual effect of reducing fibrosis and promoting wound healing by accurately preventing the delay of wound healing.

Despite the progress of studies such as the above, metformin’s effect on the burn wound microenvironment has not been reported. Our motivation for conducting these studies was related to our previous studies investigating the effects of metformin on fibrosis and inflammation.

During the inflammatory stages of wound healing, macrophages express IL-1β, IL-6, and TNF-α, which exert pro-inflammatory and bactericidal effects. Moreover, it can recruit more monocytes and exacerbate the inflammatory response of macrophages by producing potent chemokines (CCL2). With the resolution of inflammation, fibroblasts and myofibroblasts differentiate from them and secrete large quantities of ECM proteins, such as Col-I, Col-III, and Fn, play an active role in the healing process, and have an impact on the healing outcome. Myofibroblasts, for example, can control wound contraction via the ECM, affecting the “tension balance” of the wound and thus the degree of fibrosis, though excessive fibrosis frequently results in hyperplastic scarring. In our study, we were able to verify that an intermittent hypodermic injection of metformin (100 mM) reduced collagen I deposition and decreased the relative mRNA expression of IL1β and CCL2 in vivo. However, we did not find similar differences in the expression of collagen III and fibronectin, to which the timing and duration of the metformin intervention, among other factors, may have contributed. Therefore, our preliminary in vivo experiments demonstrated that metformin may have antifibrotic and anti-inflammatory effects on the skin of burned rats.

Networks of growth factors and hormones are well known to be responsible for the initiation and maintenance of fibrotic responses in vivo. Transforming growth factor-β (TGF-β) is a broad-spectrum growth factor which regulates many fundamental biological processes, including cell growth, differentiation, adhesion, the repair of proliferating tissues, and apoptosis. TGF-β1 is a member of the TGF-β superfamily; Smad3 is an important signaling gene downstream of TGF-β1. TGF-β1 activates Smad3, which induces and promotes fibrosis in TGF-β1- and Smad3-expressing organs, particularly in the skin, liver, kidneys, and lungs. Fibroblasts are the most abundant cell types in the dermis, maintaining different extracellular matrix components, including collagen, elastin, and proteopolysaccharides. In addition, they secrete various growth factors, including TGF-β, TNF-α, other cytokines, and matrix metalloproteinases (MMPs), which directly affect the proliferation and differentiation of keratinocytes and ECM formation. 

To further investigate the relationship between AMPK and the TGF-β1/Smad3 pathway, we stimulated NIH 3T3 metformin at different concentrations and observed attenuation in the activation of the TGF-β1/smad3 pathway and downstream collagen I expression. The results showed that metformin had a negative dose-dependent regulatory impact on the TGF-β1/Smad3 pathway and downstream collagen I. To further demonstrate the relationship between AMPK and TGF-β1/Smad3, we used siRNA and shRNA to knock down AMPK. The results we obtained showed that the TGF-β1/Smad3 pathway and collage I expression were enhanced after AMPK knockdown.

Another finding is that metformin could inhibit the migration of NIH 3T3 cells by activating AMPK. The Transwell migration assay showed that NIH 3T3 cells exhibited an inhibition of migration positively correlated with metformin under different concentration stimuli. In our experiments, when siAMPK was transfected into NIH 3T3 cells, the number of migrated NIH 3T3 cells was significantly higher compared with the negative control group. 

Matrix metalloproteinases (MMPs) are a class of metal-dependent proteolytic enzymes with similar structures and functions. They are important enzymes which degrade components of the extracellular matrix (ECM), play important roles in cell migration and differentiation, repair, and tissue reconstitution, and are involved in physiological and pathological processes in various tissues and organs of the human body. Furthermore, we explored the molecular mechanisms involved and found that AMPK inhibited the migration of NIH 3T3 cells by activating the AMPK/mTOR pathway and downregulated downstream AKT, MMP2, and MMP9 expression. Furthermore, the downregulation of AMPK by siAMPK resulted in increased MMP2 and MMP9 expression, and the AMPK/mTOR pathway was found to be associated with metformin. siRNA transfection is relatively inefficient: it can only be transfected briefly and cannot be inherited from daughter cells; therefore, expression levels decrease over time and the probability of experiencing off-target effects is high [34]. To further confirm AMPK’s role in mediating cell migration, AMPK knockout was performed using shRNA. Thus, we found that shAMPK increases the expression of pmTOR, pAKT, MMP2, and MMP9. These data indicate that AMPK attenuated the migration of NIH 3T3 cells by inhibiting mTOR and downstream proteins.

Lipopolysaccharide is the main element in the cell walls of Gram-negative bacteria. In addition to its bioactivity, toxicity, and immunity, which influence wound healing and scar formation, the inflammatory response it induces has important implications for wound repair. For example, its own toxic substance, lipid A, can lead to the release of pro-inflammatory factors and increased activity of inflammatory cells. NF-κB, a nuclear protein released from activated macrophages or injured cells, activates several inflammatory responses, immune responses, and gene transcription processes of various pro-inflammatory markers, thus controlling their biosynthesis. pNF-κB stimulates the expression of cytokines and chemokines, such as IL-1α, IL-1β, IL6, CCL2, CXCL10, and CXCL11, which in turn stimulate NF-κB and further activate NF-κB, leading to a sustained or amplified inflammatory response [35]. Our study demonstrated the ability of pAMPK to downregulate pNF-κB p65 expression, resulting in reductions in IL-1β and CCL2 levels. However, the anti-inflammatory effects of pAMPK, including those that occur through inhibition of the NF-κB pathway, were blocked by the intervention of the NF-κB inhibitor QNZ. Ultimately, metformin exerts its inhibitory effect on inflammation in an NF-κB-dependent way.

Indeed, LPS has also been reported to induce apoptosis and oxidative damage in RAW 264.7 cells at high levels [36]. To study the protective effects of metformin on RAW 264.7 cells stimulated by high levels of LPS, we pretreated the cells with different concentrations of metformin for 16 h and added 10 μg/mL of LPS for 8 h; we found that metformin significantly inhibited the early apoptosis of RAW 264.7 cells induced by high levels of LPS.

Normal wound healing in organisms is regulated by complex and subtle physiological processes, involving cell migration, inflammatory response, nerve control, and angiogenesis [37]. There are four distinct and overlapping stages of wound healing: hemostasis, inflammation, proliferation, and remodeling [38]. Fibroblasts and macrophages are two of the most important factors in wound healing; therefore, their interaction deserves further study. Inflammation is involved in the development of fibrosis and tissue regeneration, but the inflammatory process is neither a prerequisite for tissue regeneration nor necessarily a cause of fibrosis. Various studies have reported that macrophage-derived TGF-β promotes fibroblast growth and collagen synthesis and induces migration [39]. Eisuke Ueshima et al. [40] found that stimulating paracrine TGF-β1 from macrophages is necessary and sufficient to increase the collagen secretion of fibroblasts.

Low concentrations of LPS promote the proliferation of HPDLF. One direct approach may be that LPS promotes intracellular gene expression in cells by activating signaling pathways, such as ephrin/Eph [41], or that LPS affects Ca^2+^ the transmembrane transport inside and outside cells [42]; another indirect approach is that LPS promotes cell proliferation by stimulating the production of bioactive factors in cells. However, with increased LPS intervention times, cell tolerance decreases, cells overmature, production capacity decreases, and the secretion of multiple enzymes, such as matrix metalloproteinases, increases, leading to cytotoxicity [43].

To further elucidate the interaction between NIH 3T3 and RAW 264.7 cells in the inflammatory microenvironment and whether metformin plays a regulatory role, we co-cultured NIH 3T3 cells with RAW 264.7 cells and compared NIH 3T3 cells with NIH 3T3 cells.

Each group was divided into LPS− Met−, LPS+ Met−, and LPS+ Met+ sub-groups by intervention. In the NIH 3T3/NIH 3T3 group, NIH 3T3 cells in the LPS+ Met− group showed a significant reduction in type I collagen expression due to prolonged LPS stimulation at high LPS concentrations. NIH 3T3 cells in the LPS+ Met+ group showed a further reduction in type I collagen expression due to the effect of metformin. However, in the RAW 264.7/NIH 3T3 group, RAW 264.7 cells in the LPS+ Met− group secreted paracrine TGF-β1 in large amounts due to LPS stimulation, which was delivered to NIH 3T3 cells in the lower chamber, resulting in a significant increase in type I collagen expression in NIH 3T3 cells; in addition, in the LPS+ Met+ group, NIH 3T3 cells were downregulated by the effect of metformin. Metformin mainly reduced collagen I expression in NIH 3T3 cells by inhibiting TGF-β1 secretion in RAW 264.7 cells in RAW 264.7/NIH 3T3 co-cultures.

It should be noted that this study has many limitations. First, because RAW 264.7 causes cell polarization in most cases after siAMPK or shAMPK, the experiments yielded contradictory results; thus, these data were not published. Second, the burn microenvironment not only includes fibroblasts and macrophages, but also T cells, endothelial cells, dendritic cells, etc. [38]. In this study, we only investigated the effect of macrophages on collagen I expression in fibroblasts co-cultured with macrophages. Moreover, the mechanism through which NIH 3T3 affects RAW 264.7 cells by secreting cytokines and chemokines during the co-culture of NIH 3T3 and RAW 264.7 cells has not been further investigated. These data are relatively limited. Third, inflammation is a double-edged sword. Normal inflammatory responses can promote wound healing, although an increase in uncontrolled inflammation could lead to hypertrophic scarring—a severe disorder leading to hyperplastic tissue formation that degrades the physical properties and physiological functions of normal tissue.

In summary, we have demonstrated that metformin has an inhibitory effect on collagen I deposition and the inflammatory response during the inflammatory and proliferative phases of burn wound healing in rats, as measured by the detection of collagen I deposition and pro-inflammatory markers in burn wounds. In subsequent in vitro assays, we confirmed that: (1) metformin inhibits the proliferation, migration, and collagen I production of fibroblast cell line NIH 3T3 through the detection of migration- and fibrosis-related molecules and other indicators; (2) metformin inhibits the proliferation of mononuclear macrophage cell line RAW 264.7 and suppresses its inflammatory response and apoptosis in response to LPS stimulation through the detection of inflammation-related molecules and other assays; and (3) the preliminary mechanism of metformin regulation in these two cell types in the inflammatory microenvironment is downregulation of collagen I production in NIH 3T3 cells by inhibition of the RAW 264.7 paracrine secretion of TGF-β1, as was determined in the cell co-culture experiments.

However, the optimal dosage, timing of dosing, duration, and frequency of administration in humans to promote burn wound healing by reducing excessive inflammatory responses and fibrosis are still unknown. Further research is therefore needed to address these issues.

## 5. Conclusions

In conclusion, metformin’s functions include the inhibition of collagen I deposition and the suppression of pro-inflammatory markers IL1β and CCL2 in burn wounds in rats. Additionally, further exploration of the mechanism revealed that metformin could inhibit the proliferation of NIH 3T3 and RAW 264.7 cells. In NIH 3T3 cells, metformin could both reduce collagen I expression by activating AMPK to inhibit the TGF-β1/Smad3 pathway and reduce the migration of NIH 3T3 cells by inhibiting the AMPK/mTOR pathway. Meanwhile, metformin could inhibit the LPS-induced apoptosis of RAW 264.7 cells and reduce the downstream expression of IL1β and CCL2 by inhibiting the NF-κB pathway. More importantly, in cell co-culture, metformin could reduce the expression of collagen I in NIH 3T3 cells through the downregulation of paracellularly secreted TGF-β1 in RAW 264.7 cells. Therefore, metformin has potential clinical value in regulating wound healing in deep burn wounds. Due to the complexity of the burn wound microenvironment, further extensive studies are still needed.

## Figures and Tables

**Figure 1 cells-11-04094-f001:**
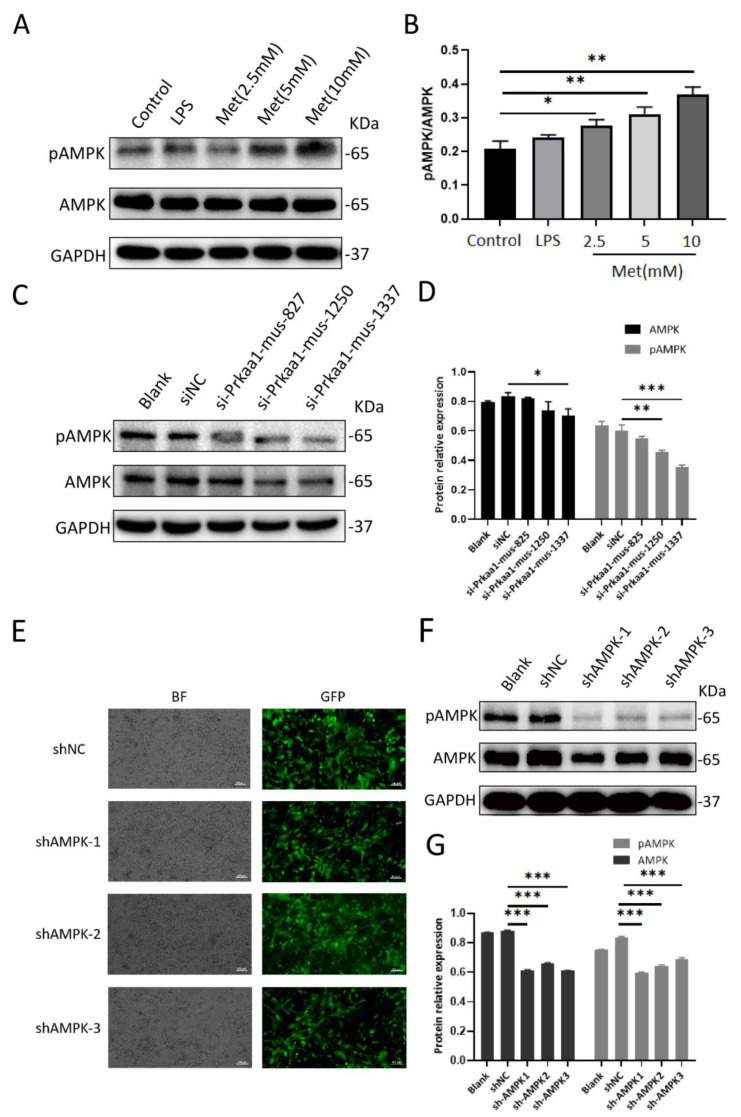
Metformin activates AMPK in a dose−dependent manner in NIH 3T3 cells. The efficiency of siRNA and shAMPK knockdown of AMPK in NIH 3T3 cells was verified by Western blotting for further experiments. (**A**,**B**) Protein expression of pAMPK and AMPK in NIH 3T3 cells with various doses of metformin treatment was detected by Western blotting. GAPDH was used as an internal control. Densitometry analysis of the Western blots demonstrated pAMPK/AMPK ratios. (**C**,**D**) Western blotting was used to measure the protein expression levels of pAMPK and AMPK in the Blank, siNC, si−Prkaa1−mus−827, si−Prkaa1−mus−1250, and si−Prkaa1−mus−1337 groups. Densitometry analysis of Western blotting results. (**E**) AMPK knockdown lentiviruses with enhanced green fluorescent protein were successfully transfected into NIH 3T3 cells. (Above, left) Small hairpin negative control group in bright field. (Above, right) Small hairpin negative control in fluorescent light. (Second row, left) Small hairpin AMPK group 1 in bright field. (Second row, right) Small hairpin AMPK group 1 in fluorescent light. (Third row, left) Small hairpin AMPK group 2 in bright field. (Third row, right) Small hairpin AMPK group 2 in fluorescent light. (Below, left) Small hairpin AMPK group 3 in bright field. (Below, right) Small hairpin AMPK group 3 in fluorescent light. BF, bright field; GFP, green fluorescent protein. Scale bar = 100 μm. (**F**,**G**) The protein expression levels of pAMPK and AMPK in the Blank, shNC, shAMPK1, shAMPK2, shAMPK3 groups. Statistical results using relative gray values from the blotting. All data are expressed as the means ± SDs of values from triplicate experiments. * *p* < 0.05, ** *p* < 0.01 and *** *p* < 0.001. Control: NIH 3T3 cells, LPS: NIH 3T3 cells treated with 1 μg/mL LPS for 24 h, Met (2.5 mM): NIH 3T3 cells treated with 2.5 mM metformin for 24 h, Met (5 mM): NIH 3T3 cells treated with 5 mM metformin for 24 h, Met (10 mM): NIH 3T3 cells treated with 10 mM metformin for 24 h, Blank: NIH 3T3 cells, siNC: siAMPK negative control group, si−Prkaa1−mus−827: NIH 3T3 cells transfected with si-Prkaa1-mus-827, si-Prkaa1-mus-1250: NIH 3T3 cells transfected with si−Prkaa1−mus−1250, si-Prkaa1−mus−1337: NIH 3T3 cells transfected with si−Prkaa1−mus−1337, shNC: lentivirus AMPK negative control group, shAMPK1: lentivirus AMPK knockdown group 1, shAMPK2: lentivirus AMPK knockdown group 2, shAMPK3: lentivirus AMPK knockdown group 3.

**Figure 2 cells-11-04094-f002:**
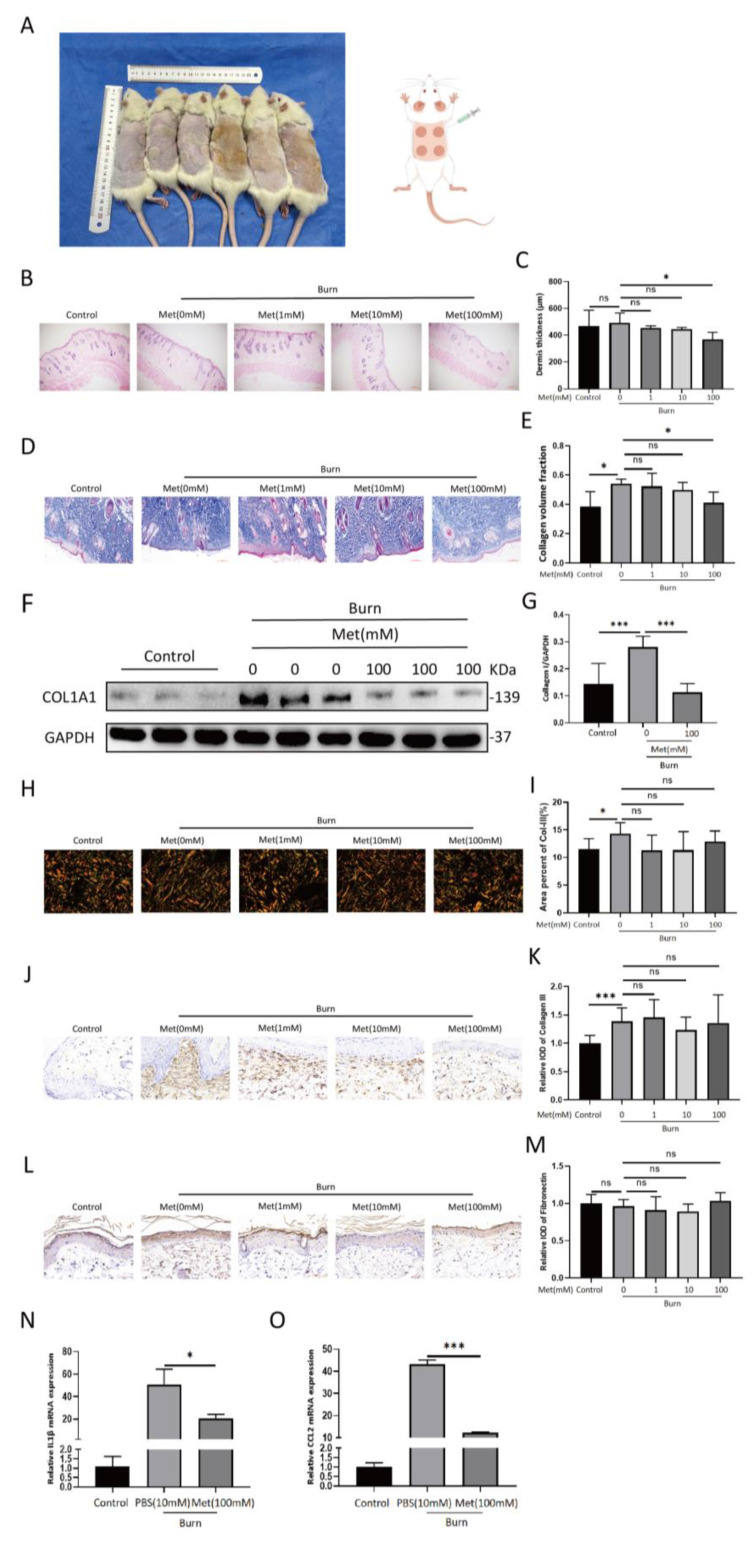
Metformin inhibits collagen I deposition and IL1β and CCL2 mRNA expression in a burn rat model. (**A**) A burn rat model is shown. Various concentrations of metformin (0, 1, 10, 100 mM) were subcutaneously injected into the dorsal burn wounds of the rats. (**B**,**C**) The rats’ skins were harvested at 14 d post−burn. Hematoxylin and eosin staining was performed on rat skin tissues under different treatments. The results of the HE stainings are shown. Scale bar = 200 μm. The statistical analysis of dermis thickness is shown in the bar graph. (**D**,**E**) Masson staining of rat skin tissues under different treatments. Statistical analysis of collagen volume fractions is shown in the bar graph. Scale bar = 100 μm. (**F**,**G**) A representative blot of three independent experiments is presented. Western blots of collagen I and GAPDH were scanned and densitometric unit ratios plotted. (**H**,**I**) Polarized light images of picrosirius red staining. Collagen III appears as fine green fibers. The areal percent of collagen III is quantified in the bar graph. Scale bar = 50 μm. (**J**,**K**) IHC for collagen III. Statistical analysis of relative ODs shown in the bar graph. Scale bar = 50 μm. (**L**,**M**) IHC for fibronectin. Statistical analysis of relative ODs shown in the bar graph. Scale bar = 50 μm. (**N**,**O**) RT−PCR analysis of IL1β and CCL2 mRNA relative expression in burned rat skin treated with or without metformin. All data are expressed as the means ± SDs of values (*n* = 12). ns: no significance. * *p* < 0.05 and *** *p* < 0.001. Control: normal rat skin, PBS (10 mM): burned rat skin treated with BPS (10 mM), Met (100 mM): burned rat skin treated with metformin (10 mM).

**Figure 3 cells-11-04094-f003:**
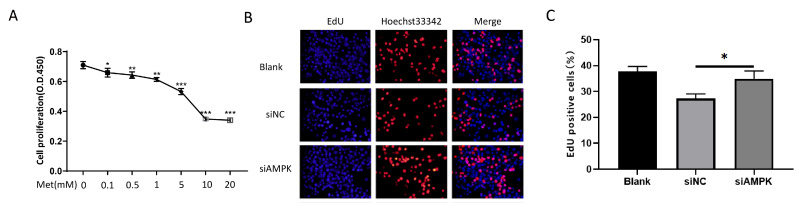
Metformin inhibits the proliferation of NIH 3T3 cells by activating AMPK. (**A**) The CCK8 test was applied to assess NIH 3T3 cells treated with metformin at various doses for 24 h. (**B**,**C**) Cell proliferation analysis by EdU incorporation assay. Transfected with siAMPK for 24 h, the cell proliferation ability of NIH 3T3 cells was measured by EdU assay. The percentages of EdU−positive cells are quantified in the bar graph. All data are expressed as the means ± SDs of values from triplicate experiments. * *p* < 0.05, ** *p* < 0.01, and *** *p* < 0.001. Blank: NIH 3T3 cells, siNC: siAMPK negative control group, siAMPK: NIH 3T3 cells transfected with si−Prkaa1−mus−1337.

**Figure 4 cells-11-04094-f004:**
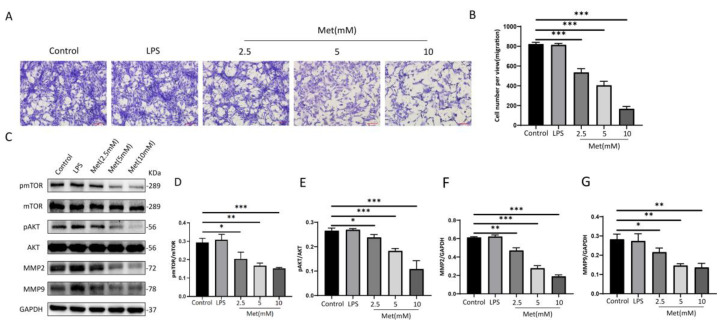
Metformin inhibits NIH 3T3 cell migration by activating AMPK/mTOR in a concentration−dependent manner. (**A**,**B**) NIH 3T3 cells in the upper chamber were incubated in the presence of different concentrations of metformin (2.5 mM, 5 mM, 10 mM) for 24 h and the number of migrating cells was observed in five random fields under a 100× microscope. Scale bar = 100 μm. Cell migration was quantified using ImageJ/Fiji software. Values represent the means ± SEs of the numbers of migrating cells from three independent experiments. (**C**−**G**) Relative band densities of pmTOR/mTOR, pAKT/AKT, MMP2, and MMP9 are quantified in the bar graphs. All the experiments were performed in triplicate and repeated on at least three occasions. Data are shown as means ± SDs. * *p* < 0.05, ** *p* < 0.01, and *** *p* < 0.001. Control: NIH 3T3 cells, LPS: NIH 3T3 cells treated with 1 μg/mL LPS for 24 h, Met (2.5 mM): NIH 3T3 cells treated with 2.5 mM metformin for 24 h, Met (5 mM): NIH 3T3 cells treated with 5 mM metformin for 24 h, Met (10 mM): NIH 3T3 cells treated with 10 mM metformin for 24 h.

**Figure 5 cells-11-04094-f005:**
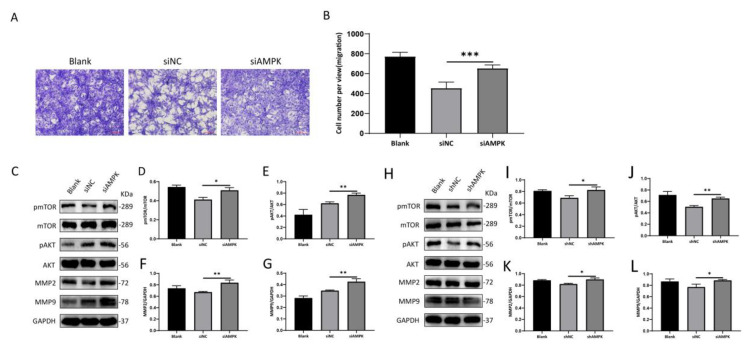
Knockdown of AMPK promotes the migration of NIH 3T3 cells. NIH 3T3 cells were incubated for 24 h. (**A**,**B**) The number of migrating cells was observed in five random fields under a 100× microscope. Scale bar = 100 μm. Cell migration was quantified using ImageJ/Fiji software. Values represent the means ± SEs of the numbers of migrating cells from three independent experiments. (**C**–**L**) Relative band densities of pmTOR/mTOR, pAKT/AKT, MMP2, and MMP9 are quantified in the bar graphs. All the experiments were performed in triplicate and repeated on at least three occasions. Data are shown as means ± SDs. * *p* < 0.05, ** *p* < 0.01, and *** *p* < 0.001. Blank: NIH 3T3 cells, siNC: siAMPK negative control group, siAMPK: NIH 3T3 cells transfected with si−Prkaa1−mus−1337, shNC: lentivirus AMPK negative control group, shAMPK: lentivirus AMPK knockdown group 1.

**Figure 6 cells-11-04094-f006:**
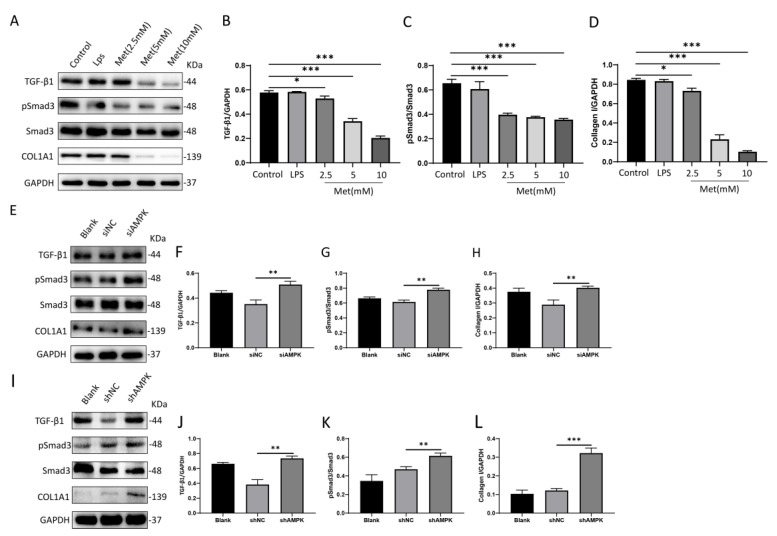
Metformin inhibits the TGF−β1/Smad3 pathway in NIH 3T3 cells by activating AMPK, which in turn decreases the expression of collagen I. (**A**–**D**) NIH 3T3 cells were treated with metformin at doses of 2.5 mM, 5 mM, and 10 mM in a high-glucose medium for 24 h. Protein levels of TGFβ-1, phosphorylated Smad3 (pSmad3), Smad3, and collagen I were determined by Western blotting and analyzed by densitometry. The representative data are shown. (**E**–**L**) Representative Western blots and quantitative results for TGFβ-1, phosphorylated Smad3 (pSmad3), Smad3, and collagen I proteins. All data are expressed as the means ± SDs of values from triplicate experiments. * *p* < 0.05, ** *p* < 0.01, and *** *p* < 0.001. Control: NIH 3T3 cells, LPS: NIH 3T3 cells treated with 1 μg/mL LPS for 24 h, Met (2.5 mM): NIH 3T3 cells treated with 2.5 mM metformin for 24 h, Met (5 mM): NIH 3T3 cells treated with 5 mM metformin for 24 h, Met (10 mM): NIH 3T3 cells treated with 10 mM metformin for 24 h, Blank: NIH 3T3 cells, siNC: siAMPK negative control group, siAMPK: NIH 3T3 cells transfected with si−Prkaa1−mus−1337, shNC: lentivirus AMPK negative control group, shAMPK: lentivirus AMPK knockdown group 1.

**Figure 7 cells-11-04094-f007:**
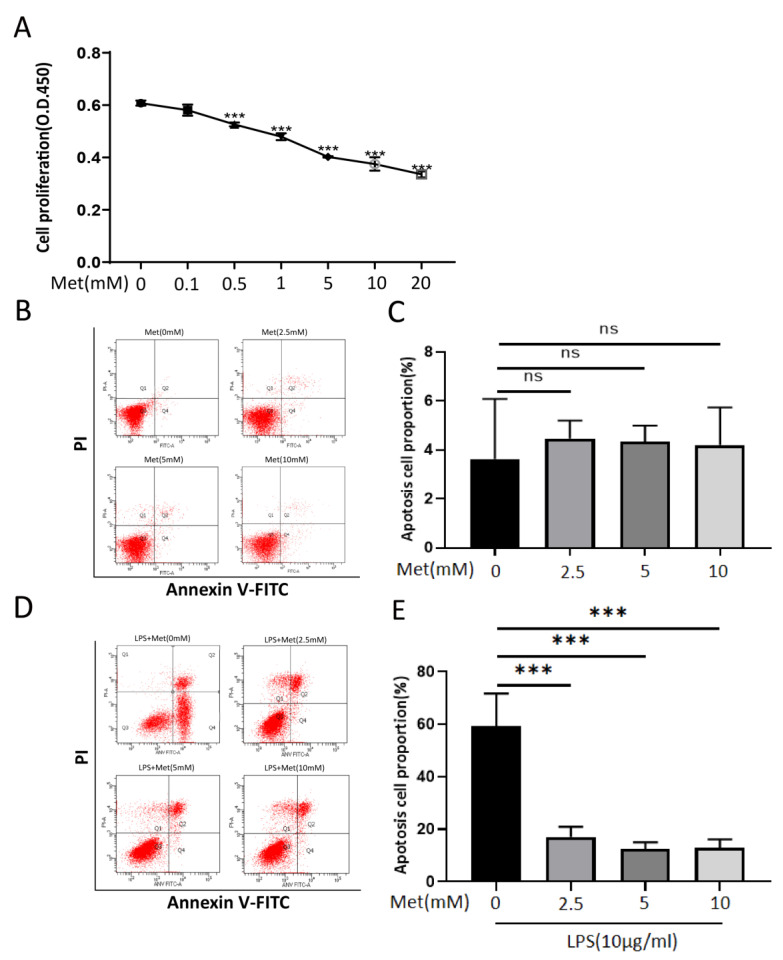
Metformin inhibits the proliferation of RAW 264.7 cells. Metformin protects against apoptosis of RAW 264.7 cells induced by high concentrations of LPS. (**A**) Cell viability was assessed by CCK8 assay. RAW 264.7 cells were treated with different concentrations of metformin (0, 0.1, 0.5, 1, 5, 10, 20 mM) for 24 h. (**B**,**C**) Effects of metformin at various dose on apoptosis of RAW 264.7 cells. No significant differences were detected by quantification via Annexin V-FITC/PI flow cytometry. (**D**,**E**) Effects of pretreatment of RAW 264.7 cells with different concentrations of metformin on apoptosis induced by LPS (10 μg/mL). The percentages of apoptotic cells are quantified in the bar graph. All experiments were repeated three times and means ± SDs of the data are presented. ns: no significance, *** *p* < 0.001. Met (0 mM): NIH 3T3 cells treated without metformin for 24 hMet, Met (2.5 mM): NIH 3T3 cells treated with 2.5 mM metformin for 24 h, Met (5 mM): NIH 3T3 cells treated with 5 mM metformin for 24 h, Met (10 mM): NIH 3T3 cells treated with 10 mM metformin for 24 h, LPS + Met: NIH 3T3 cells treated with metformin for 16 h, then stimulated with the addition of LPS (10 μg) for 8 h.

**Figure 8 cells-11-04094-f008:**
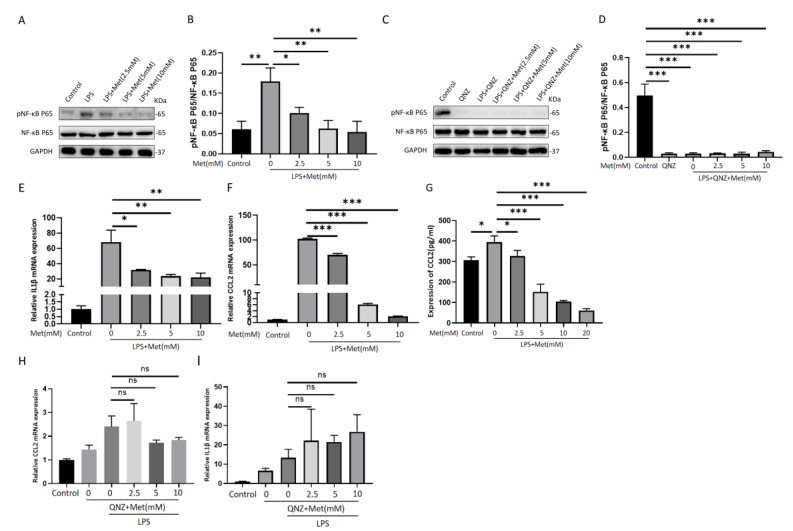
Metformin−mediated reduction in phospho−NF−κB and IL1β and CCL2 is abolished by treatment with QNZ, an inhibitor of NF−κB. (**A**,**B**) Western blots showing the effect of metformin pretreatment on the expression of pNF−κB and NF−κB in RAW 264.7 cells. Densitometry analysis of Western blotting results. (**C**,**D**) Protein expression of pNF−κB and NF−κB in RAW 264.7 cells after QNZ and metformin preintervention is quantified in the histogram. (**E**,**F**) Changes in mRNA expression of IL1β and CCL2 in RAW 264.7 cells stimulated by LPS after pretreatment with different doses of metformin. (**G**) ELISA showing the effect of metformin on CCL2 expression in RAW 264.7 cells. (**H**,**I**) Changes in mRNA expression of IL1β and CCL2 in RAW 264.7 cells stimulated by LPS after pretreatment with QNZ and various doses of metformin. All experiments were repeated three times and means ± SDs of the data are shown (*n* = 3). ns: no significance. * *p* < 0.05, ** *p* < 0.01, and *** *p* < 0.001. Control: RAW 264.7 cells, QNZ: RAW 246.7 cells pretreated with QNZ, LPS + Met (mM): pretreatment with metformin followed by LPS intervention, LPS + QNZ + Met (mM): pretreatment with metformin and QNZ, followed by LPS intervention.

**Figure 9 cells-11-04094-f009:**
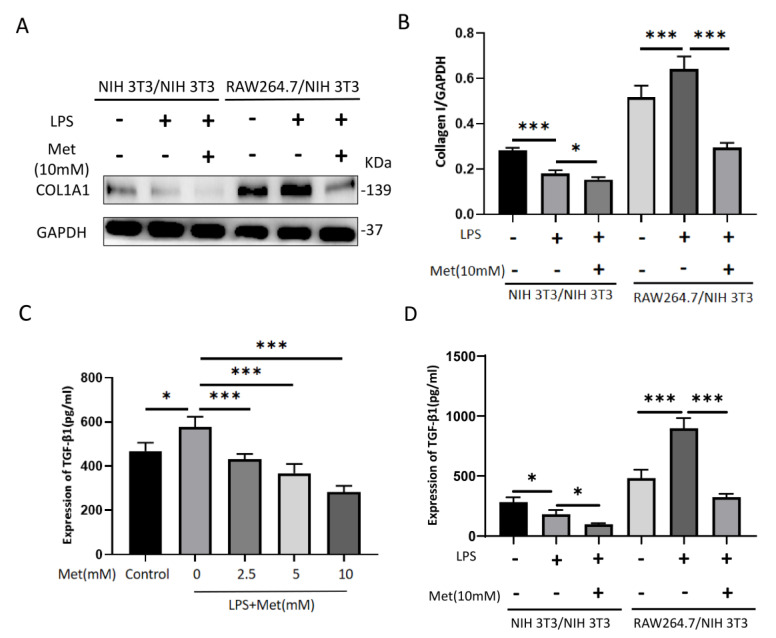
Metformin inhibits the expression of collagen I in NIH 3T3 cells in a co−culture system by suppressing TGF−β1 paracrine secretion in RAW 264.7 cells. (**A**,**B**) Representative Western blot bands of collagen I in each co−culture group. GAPDH was used as a loading control. Semiquantitative analysis of the expression of collagen I in each group. (**C**) ELISA assay of TGF−β1 expression in RAW 264.7 cells after 6 h of LPS (1 μg/mL) stimulation with or without different concentrations of metformin (0−10 mM) pretreatment for 18 h. (**D**) ELISA assay was performed to detect the expression of TGF−β1 in each cell co−culture group. Data indicate means ± SDs of three independent experiments. * *p* < 0.05 and *** *p* < 0.001. NIH 3T3/NIH 3T3 (LPS− Met−): NIH 3T3 cells and NIH 3T3 cells were co−cultured for 48 h without LPS or Met, NIH 3T3/NIH 3T3 (LPS+ Met−): NIH 3T3 cells and NIH 3T3 cells were co−cultured for 48 h with 1μg/mL of LPS, NIH 3T3/NIH 3T3 (LPS+ Met+): NIH 3T3 cells and NIH 3T3 cells were co−cultured for 48 h with 1 μg/mL of LPS and 10 mM of Met, RAW 264.7/NIH 3T3 (LPS− Met−): RAW 264.7 cells and NIH 3T3 cells were co−cultured for 48 h without LPS and Met, RAW 264.7/NIH 3T3 (LPS+ Met−): RAW 264.7 cells and NIH 3T3 cells were co−cultured for 48 h with 1 μg/mL of LPS, RAW 264.7/NIH 3T3 (LPS+ Met+): RAW 264.7 cells and NIH 3T3 cells were co−cultured for 48 h with 1 μg/mL of LPS and 10 mM of Met.

**Table 1 cells-11-04094-t001:** Mean inhibitory concentrations (IC50) of metformin in NIH 3T3 and RAW 264.7 cell lines (*n* = 3; means and confidence intervals (95%)).

	IC50 (mM)	95% IC50 (mM)
NIH 3T3	2.71	1.02~3.282
RAW 264.7	2.15	1.32~2.98

**Table 2 cells-11-04094-t002:** Sequences of synthetic siRNAs used for transient transfection.

Name		Sequence (5′ ~ 3′)
Prakaa1-mus-827	Sense	GGGAACACGAGUGGUUUAATT
	Antisense	UUAAACCACUCGUGUUCCCTT
Prakaa1-mus-1250	Sense	GCCGACCCAAUGAUAUCAUTT
	Antisense	AUGAUAUCAUUGGGUCGGCTT
Prakaa1-mus-1337	Sense	GCGUGUACGAAGGAAGAAUTT
	Antisense	AUUCUUCCUUCGUACACGCTT
Negative control	Sense	UUCUCCGAACGUGUCACGUTT
	Antisense	ACGUGACACGUUCGGAGAATT

**Table 3 cells-11-04094-t003:** Sequences of short hairpin RNA used for lentiviral shRNA assay.

Name		
shAMPK1	Top strand	GATCCGCAGAAGTCATTTCAGGAAGATTGTATTCAAGAGATACAATCTTCCTGAAATGACTTCTGTTTTTTG
	Bottom strand	AATTCAAAAAACAGAAGTCATTTCAGGAAGATTGTATCTCTTGAATACAATCTTCCTGAAATGACTTCTGCG
shAMPK2	Top strand	GATCCGTCTCTTTCCTGAGGACCCATCTTATTTCAAGAGAATAAGATGGGTCCTCAGGAAAGAGATTTTTTG
	Bottom strand	AATTCAAAAAATCTCTTTCCTGAGGACCCATCTTATTCTCTTGAAATAAGATGGGTCCTCAGGAAAGAGACG
shAMPK3	Top strand	GATCCGAGCAATCAAGCAGTTGGATTATGAATTCAAGAGATTCATAATCCAACTGCTTGATTGCTTTTTTTG
	Bottom strand	AATTCAAAAAAAGCAATCAAGCAGTTGGATTATGAATCTCTTGAATTCATAATCCAACTGCTTGATTGCTCG
shNC	Top strand	GATCCGTTCTCCGAACGTGTCACGTAATTCAAGAGATTACGTGACACGTTCGGAGAATTTTTTC
	Bottom strand	AATTGAAAAAATTCTCCGAACGTGTCACGTAATCTCTTGAATTACGTGACACGTTCGGAGAACG

**Table 4 cells-11-04094-t004:** Rat primers for quantitative real-time PCR analysis.

Name		Sequence (5′~3′)	Concentration
CCL2	Forward primer	CTCACCTGCTGCTACTCATTCACTG	150 nM
	Reverse primer	CTTCTTTGGGACACCTGCTGCTG	150 nM
IL1β	Forward primer	AATCTCACAGCAGCATCTCGACAAG	150 nM
	Reverse primer	TCCACGGGCAAGACATAGGTAGC	150 nM
GAPDH	Forward primer	GACATGCCGCCTGGAGAAAC	150 nM
	Reverse primer	AGCCCAGGATGCCCTTTAGT	150 nM

**Table 5 cells-11-04094-t005:** Mouse primers for quantitative real-time PCR analysis.

Name		Sequence (5′~3′)	Concentration
CCL2	Forward primer	TTTTTGTCACCAAGCTCAAGAG	150 nM
	Reverse primer	TTCTGATCTCATTTGGTTCCGA	150 nM
IL1β	Forward primer	TCGCAGCAGCACATCAACAAGAG	150 nM
	Reverse primer	AGGTCCACGGGAAAGACACAGG	150 nM
GAPDH	Forward primer	GGTTGTCTCCTGCGACTTCA	150 nM
	Reverse primer	TGGTCCAGGGTTTCTTACTCC	150 nM

## Data Availability

The data presented in this study are available on request from the corresponding author upon reasonable request.

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
