# Peer review of "Metformin Improves Burn Wound Healing by Modulating Microenvironmental Fibroblasts and Macrophages"

_cells, 2022, doi:10.3390/cells11244094_

Round 1

Reviewer 1 Report

* In the Abstract, please specify in the second sentence how the metformin was administered. 

* Is it a good thing that metformin inhibits proliferation of fibroblasts and macrophages locally? 

* Many things act through mTOR and Smad 3. How were these other things controlled?  Could you address this with other gene silencing?  

*  The Introduction is a bit choppy, and in places difficult to understand. It seems to be missing a clause or sentence at the end of the third paragraph (or did it carry on in the next paragraph?). In particular, it seems that more introduction on the mechanism of metformin effects in general and in particular glucose metabolism are needed.  For instance, why does changing the 'cellular energy status' induce better glucose control?

*  The effects of metformin on the other systems listed also do not have a clear description of exactly how these effects occur.  Is it all through mitochondria?

*  With these effects clearly described, how did these lead you to think that metformin would have anti-inflammatory effects on fibroblasts?  The story line is not clear,...

*  Unfortunately, the study design does not really lead to 'mechanism' but instead to associated factors for the in vivo experiments, and these mostly from known mediators.  How exactly is IL-1beta affected? 

*  The distribution of the metformin in the animal experiments is not clear.  You have a range of doses, but only 12 rats; did each rat receive different doses? How exactly? How far apart were the injections? Any carry-over or distributive effects? The description should be in enough detail that it could be repeated reasonably by a reader.

*  Why did you choose to dose every 48 hours? The published half-life of metformin is 6 hours,...

*  AMPK silencing RNAs were used.  Why only for this actor in that pathway?  What about molecules proximal and distal in the pathway?  Why only in the fibroblasts and not in the macrophages?

*  For Figure 1, what you have shown is that siRNAs for AMPK do in fact silence AMPK.  What about after metformin treatment? 

* How exactly are IL1beta and CCL2 related to AMPK?  Perhaps a cartoon demonstrating the pathway would be of benefit.  You do demonstrate conclusively that AMPK is involved in fibroblast proliferation and migration, but how is this related to IL-1 and CCL2?

* Human scarring is generally a function of collagen dys-array after wound healing. Did you consider doing this after healing or at least discussing it? Also, we don't have the westerns for the lower doses in the fibure.

* In general, this is a reasonably well done experiment and manuscript, but it only tells part of the story. Many things were measured, but we don't see anything that wasn't changed or things that might be in the periphery. I can't name any specifically, but showing something that was NOT changed for positive control would make this more attractive.  Perhaps STAT pathways? 

* The Discussion is done well, but it is missing a summary of what this adds to the literature. What exactly does metformin do in wound healing in a single sentence and how does this data support the contention. Further, what was not measured that might have consequence. 

Reviewer 2 Report

Congratulations to this very interesting manuscript on an important topic in burn treatment.

I would allow myself the following questions for further clarification and improvement of the manuscript:

- the abstract would profit from a more concise structure including a (very short) introduction of the in-vitro cell types (NIH/RAW) for better understanding

- Both methods and results should provide more information (i.e. pictures, graphs) on the state of the wounds after 14 days: were the burn wounds completely healed? What was the time to wound/skin closure? Was scar formation visible macroscopically? Any occurences of wound infection or delayed closure inter/intraindividually?

- What kinds of topical dressings were used for wound coverage during observation?

- Which area were the day 4 in vivo samples taken from - the wound bed or the edge of the wound/scar?

- Figure 2 shows a decrease of total dermal thickness with a comparable collagen volume fraction, and only differences in collagen I expression. Please provide data on other collagen types such as collagen 3. Have the effects of metformin in the in-vitro assays also been studied on collagen III and VII? Please elaborate and discuss

- Figure 4 is very busy and might profit from omitting the western blot graphs

- english grammar editing is advised to improve readability and clarity of the manuscript

Round 2

Reviewer 1 Report

I understand your response to defining collagen III expression, though it is disappointing that this analysis was not done.  Same for collagen VII.  I presume you still have the samples, and to fully define scarring in this rodent model for application to human disease, unfortunately, this should still be done before the paper is published.
